# Experimental Research on Variable Parameter Forming Process for Forming Specimen of TC4 Titanium Alloy by Selective Laser Melting

**DOI:** 10.3390/ma15186408

**Published:** 2022-09-15

**Authors:** Yude Liu, Yusheng Zhou, Wentian Shi, Jian Han, Donglei Ye, Yufan Han

**Affiliations:** School of Artificial Intelligence, Beijing Technology and Business University, Beijing 100048, China

**Keywords:** selective laser melting, variable parameter forming process, surface morphology, tensile properties, microstructure

## Abstract

To optimize the microstructure and properties of TC4 specimens formed by selective laser melting (SLM), the test program of formed specimens by the variable parameter forming process (VPFP) was designed based on the quantitative parameter forming process (QPFP). The purpose of this study is to explore the influence of the VPFP on the surface morphology, tensile properties, and microstructure of the specimens. The test results show that the surface morphology and tensile properties of the specimens were better formed by the VPFP. The internal holes of the specimens formed by the VPFP were small in volume and occupied a relatively small proportion, and the density could reach 99.7%. When the laser power was 300 W–260 W and equally divided into six hierarchies, the tensile strength could reach 1185.214 MPa by VPFP, but the elongation had no obvious change. The number of secondary acicular martensite *α’* phases was decreased in the microstructure of the specimens formed with VPFP. With the superposition of the hierarchy, the length of the primary acicular martensite *α’* phase became shorter, the width became larger, and the width of the columnar crystal *β* phase became smaller. The VPFP is used to change the inherent method of forming specimens with the same parameters, which provides a new idea for SLM-forming structures; the test provides data and yields a theoretical research basis for forming the specimens process method.

## 1. Introduction

Selective laser melting (SLM) additive manufacturing technology is a discrete and stacking concept, which uses the laser beam to melt metal powder layers, manufacturing the 3D solid model layer-by-layer from bottom to top to form complex structure metal parts [1,2,3]. SLM has a broad application prospect in aerospace, weapons and equipment, automobile, mold, biomedicine, and other fields [4]. However, due to the complexity of the SLM forming process, it involves materials with metallurgical, physical, chemical, and thermo-mechanical properties. The characteristics of point-by-point, line-by-line, and surface-by-surface forming lead to inevitable defects inside the alloy specimens. Some defects significantly reduce the mechanical properties of the alloy specimens. It is a key problem to avoid defects and improve the mechanical properties of structures in SLM technology, which has become a hotspot in recent research [5,6,7,8,9,10,11,12,13,14].

Currently, researchers have carried out relevant studies on the formation of internal defects in SLM specimens and their influence on mechanical properties. Kasperovich et al. [15] optimized process parameters to process specimens with TC4. The experimental results showed that the appropriate process parameters can significantly reduce the porosity of the sample, and the minimum porosity is 0.05%. When the volume energy density is very high, there are holes produced by the keyhole effect. When the volume energy density is insufficient, there are the defects of being slender and perpendicular to the forming direction caused by the incomplete melting of metal powder. Benedetti et al. [16] studied the tensile properties of TC4 specimens formed by SLM. With certain process parameters, the fatigue strength of the TC4 specimen is only 20% of the tensile strength of machined titanium alloy, and this fatigue strength of the sample is related to the maximum defect size in the sample. DebRoy et al. [17] studied the volatilization of alloying elements during additive manufacturing of engineering alloys and reviewed the formation of defects, such as holes, unfused powder, and cracks. The residual stresses of SLM specimens were measured by Vrancken et al. [18] It proved that the residual stresses of SLM specimens are generally positively correlated with the thermal gradient during deposition. Wei et al. [19] predicted the actual printing process according to the model during the process of additive preparation. The application of the migration model in defect formation and residual stress was reviewed by Zhao et al. [20], who observed the details of the formation process of small hole defects under the condition of high power and low scanning speed laser melting by using high-speed X-ray imaging. It was found that the boundary of the small hole in the power and speed space was sharp and smooth, and there was little change between the substrate and the powder bed. Lu et al. [21] studied the preparation of the TC4 titanium alloy by a combination of LSP and SLM and conducted the analysis on the phase composition, residual stress distribution, mechanical properties, and microstructure of the specimens parallel to and perpendicular to the forming direction. It presented that the mixed additive preparation technology could realize the efficient and high-quality integrated preparation of the practical alloy.

The experimental research mentioned above used different methods to improve the high-quality performance of SLM-formed specimens, which greatly improved the tensile strength of the specimens. However, SLM-formed specimens usually used the same number of parameters to form the specimens. The influence of the heat transfer range of laser energy on the mechanical properties of the formed specimens was not considered during the forming process, with the increase of the height of the deposition layer. In this manuscript, the experimental specimens were equally stratified, and the parameters decreased layer-by-layer from the initial layer by the variable parameter forming process (VPFP). The specimens were formed from different bottoms to explore the differences between the specimens formed by the quantitative parameter forming process (QPFP). The microstructure and tensile strength of the specimens formed by the two processes were compared, the tensile strength of the specimens was observed, and the optimal mechanical properties were sought within a certain range of parameters. In this experiment, the thickness of the specimen was 0.9 mm, the VPFP was used to form the specimen, and six layers, at most, were used to form the thin samples by SLM. It provides a new idea for SLM forming structures and provides certain data and a theoretical research basis for a forming specimen process method.

## 2. Experimental Procedure

In this study, the material was Ti-6Al-4V (TC4) titanium alloy powder (prepared by gas atomization). Figure 1a shows the characteristics of TC4 titanium alloy powder particles by scanning electron microscope (SEM). Figure 1b presents particle size distribution of the power, with range of 15–53 μm. Additionally, the average particle size was 35 μm. It can be seen from Figure 1 that TC4 titanium alloy powder has high sphericity, and no impurity adhesion on the powder surface. As shown in Table 1, the chemical composition of the alloy powder was determined by inductively coupled plasma optical emission spectrometry (ICP-OES). The substrate material was TC4 to ensure the combination of materials and substrate. In order to ensure the powder drying before carrying out the experiment, TC4 titanium alloy powder was placed in a vacuum oven for drying treatment to remove the moisture that may be adsorbed on the surface.

In this study, RENISHAW AM400 (Figure 2a) (Renishaw plc, London, UK) was used to build TC4 specimens. This SLM equipment was equipped with an Nd: YAG laser with a wavelength of 1075 nm and a laser beam diameter of 70 μm. The maximum power was 400 W in a continuous pulsed laser mode. The maximum volume of the parts were processed with 250 mm × 250 mm × 300 mm. The working area needed to be filled with argon as a protective gas to provide a closed environment, and the oxygen concentration had to be less than 200 ppm.

The tensile experiment was carried out with the mechanical tensile machine (Instron 5966, The Northwest Chemical Research Institute Co. LTD, China, Figure 2b), and the microstructure of the specimen was observed by the super depth of field microscope (KEYENCE vhx-600e, Japan, Figure 2c) and scanning electron microscope (phenom XL, Netherlands, Figure 2d). Kroll reagent was used to corrode the surface of the polished specimen. Firstly, the metal surface was wiped by absorbent cotton stained with corrosive liquid for 2 min. Secondly, when the metal surface was dark, the residual Kroll reagent was washed off on the surface of the specimen with ionic water. Finally, the surface was rinsed with by alcohol and then blow-dried. The automatic stereoscopic zoom microscope (ASI SZ-2000 USA) was used to measure the roughness of the samples.

During the process of SLM specimens, the local transferring of energy caused a large temperature gradient, diffusing along the energy center to the surroundings. The thermal conductivity of loose metal powder was lower than that of the substrate and the formed specimen, resulting in large residual stress during the energy transfer process [22,23]. With the increase in the forming height of the specimen, the solidification shrinkage behavior of the latter deposition layer was bound by the solidification of the previous deposition layer under the same process parameters. The temperature gradient in the metal powder propagation was increased, compared with the previous deposition layer (Figure 3a,b). With the area of the external conduction temperature gradient increasing, the cooling rate slowed down, resulting in the residual stress of the latter deposition layer being higher than that of the previous deposition layer: the residual stress had an important influence on the mechanical properties of SLM-formed alloy specimens. The energy of SLM-formed specimens is mainly characterized by bulk energy density, which can be expressed as follows:(1)$$E=\frac{p}{vht}$$
where *E* is the volume energy density, *p* is the laser power, *v* is the laser scanning speed, *h* is the scan line spacing, and *t* is the layer thickness.

According to the above analysis and the experimental results of the forming specimens, the point distance of 35 μm, exposure time of 80 s, scan line spacing of 0.07 mm were chosen in the experiments. Laser power and experimental parameters are shown in Table 2.

The experimental data of specimens 1–8# were referred to by Shi W et al. [24]. For specimens 9–20#, as seen in the above experimental data, the laser power was selected at 250 W–300 W. Additionally, the specimens were divided into six hierarchies (the hierarchy refers to the number of layers divided and aggregate thickness refers to the thickness of the hierarchy), from the first hierarchy to the sixth hierarchy, and corresponding experiments were conducted on each different hierarchy. The laser power between 250 W and 300 W was equally distinguished with the change of hierarchy. Due to the characteristics of the machine, the laser power used integers, and the laser power was rounded off after being equally distinguished.

Figure 4a presents the geometrical size of the tensile specimen. Additionally, Figure 4b presents the actual drawing of tensile parts, with a thickness of 0.9 mm. The support structure was a lattice structure with a height of 0.5 mm. Different process parameters were chosen for different hierarchies; the upper layer with a lower volume energy density than the lower layer. Figure 4c shows the distribution of the volume energy density of the specimens with six layers and a laser power of 300 W–250 W, with variable parameters that are equally divided.

## 3. Results and Discussion

### 3.1. Density and Surface Roughness

#### 3.1.1. Density Analysis

Density is one of the standard characterizations used to judge the quality of SLM-forming specimens, which reflects the degree of defects, such as holes and cracks. The Archimedes drainage method was used to obtain the density, and the density of the specimens was measured three times. The density measurement results of the specimens are shown in Figure 5. The calculation formula can be expressed as follows:(2)θ={M/[(M1−M0)ρ0]ρ1}×100%
where *θ* is density, *M* is TC4 specimen quality, *M*_1_ is TC4 specimens immersed in water and total mass of water, *M*_0_ is the quality of water, *ρ*_0_ is the density of water, and *ρ*_1_ is the standard density of TC4 titanium alloy.

The density of the specimen is shown in Figure 5. According to the experimental parameters in Table 2, when the specimens were 2–8#, the density decreased when the laser power was decreases. Under the process parameters of this experiment, the laser power was the only factor of volume energy density. When the laser power was 200 W, the density was 98.6%. Due to the low-volume energy density, the molten pool had a low liquidus temperature, high viscosity, and poor spread ability, resulting in the incomplete overlap of the molten pool. At the same time, the unmelted powder hindered the fusion of the powder layers and produced process holes (Figure 6a). The density of specimens lower than that of sample 2#, and the deposited sample with a high-volume energy density, produced thermal stress cracks (Figure 6b). At the same layer height, the higher the average number of hierarchies was, the higher the density of the specimen was, with the highest density being 99.7%. Under the same layer and with specimens 12–15# and 16–19#, the density of the specimen formed with the same volume energy density was greater than that of the specimen with a decreasing volume energy density. The main reason for this was that the density decreases with the QPFP by the decreasing volume energy density of specimens 2–8# because the density of the specimen formed by the QPFP decreases with the decrease of laser power. As for the specimen formed by the VPFP, the surface-forming quality of the front deposition layer affected the surface-forming quality of the next deposition layer. The fewer defects there were, the less accumulated surface defects of the post-deposition layer there were, and the forming quality was improved. Therefore, the density of the VPFP was better than that of the QPFP.

#### 3.1.2. Surface Roughness Analysis

The surface roughness of specimens formed by the VPFP and QPFP is shown in Figure 7. When the laser power of specimens 3#–7# decreases, the surface roughness was increased. The top laser power of specimens 12#, 13#, 14#, 15#, and 9# were the same as that of specimens 3–7#. When laser power was decreasing, the surface roughness of the specimen formed by the VPFP was increasing; however, it was lower than the surface roughness of the specimen formed by the QPFP. Energy absorbed by powder was proportional to the laser power in SLM, but a higher volume energy density increased the wettability of the molten pool, the lower volume energy density led to insufficient melting of the powder, and the surface roughness was increased. During the specimen-forming process, the affected factors of density were holes and cracks. With the increase of hierarchy, the fluidity of melt was more and more unstable, resulting in more holes, the formation of the next deposition layer, and a difference in powder thickness. Under the same volume energy density, the melting degree of powder in the hole part was lower than that in the non-hole part, and the defects accumulated in the upper surface, which reduced the surface-forming quality of the specimen. During the VPFP, cracks and holes were reduced, so the surface quality was improved due to the relatively good surface quality and the relatively regular melt flow.

Through the analysis of specimen density and surface roughness, it can be seen that the surface-forming quality of the VPFP was higher than that of the QPFP. In the experimental study of this manuscript, the maximum density of the forming specimen formed by the VPFP could reach 99.7%. By comparing the QPFP with a laser power of 290 W, 280 W, 270 W, 260 W and 250 W and the VPFP with a laser power of 300 W–250 W, evenly divided into six hierarchies, forming a layered laser power of 290 W, 280 W, 270 W, 260 W, 250 W surface roughness, it can be concluded that the VPFP improved the surface roughness.

### 3.2. Mechanical Properties and Fracture Morphology Analysis

#### 3.2.1. Mechanical Properties Analysis

The tensile strength and elongation of specimens formed by the QPFP and VPFP are shown in Figure 8. It can be seen from the figure that the tensile strength of the specimen formed by the QPFP decreased with the decrease in laser power, with a maximum tensile strength of 987.958 MPa (1#) and a maximum elongation of 5.754% (4#). Additionally, the tensile strength of the specimen formed by the VPFP increased with the increase in hierarchy: the maximum tensile strength was 1159.302 MPa (9#) and the maximum elongation was 6.812% (10#). In this experiment, the tensile strength of specimens formed by the VPFP was higher than that of specimens formed by the QPFP, but the difference of elongation was not obvious between these two processes. During the process of forming the specimen by the VPFP, the laser powers decreased regularly with the increase of the deposition layer in the specified range. With the decrease in volume energy density, the increase of the deposition layer led to an increase in temperature diffusion in the powder. The thermal conductivity of the powder was lower than that of the formed specimen, resulting in heat accumulation in the forming layer, residual stress, and a reduction in the tensile strength of the specimen. By comparing the tensile strength of the QPFP and VPFP, it can be seen that the VPFP could improve the tensile properties of specimens.

In order to further explore the difference between the tensile properties of the specimens formed by the QPFP and the VPFP, the change trends in tensile properties of the specimen with a laser power of 300 W and a laser power of 300 W–250 W divided into 2–6 hierarchies are shown in Figure 9. Through the comparison of the tensile properties between the hierarchies and the difference between the VPFP and the QPFP, disparity of tensile properties can be analyzed. It can be seen from the figure that, from the second layer, the tensile strength of the specimen formed by the VPFP was higher than the tensile strength of the specimen formed by the QPFP. Additionally, the tensile strength change curve of the QPFP showed a trend in linear function. The difference between the tensile strength of the row layer and the previously deposited layer was within a certain range. The VPFP resulted in the difference of tensile strength between the formed row layer and the previously deposited layer, which increased first and then decreased. The laser power of 300 W–250 W was divided into six hierarchies to form the specimen. When the laser power of 250 W formed the specimen deposition layer, due to the low-volume energy density, the tensile properties of the formed specimens were reduced. When the laser power of the fifth layer was 260 W, the tensile strength difference between the VPFP and the QPFP was the largest. The laser power range of the VPFP was adjusted. The laser power of 300 W–260 W was evenly divided into six hierarchies for the second experiment, and the tensile strength of the specimen reached 1185.214 MPa.

#### 3.2.2. Analysis of Fracture Morphology

The tensile fracture morphology of specimens formed by different processes is shown in Figure 10. The fracture morphology of the specimen was a quasi-cleavage fracture. Figure 10a shows that the fracture morphology was partially formed by dimple morphology, mainly composed of an irregular quadrilateral. In Figure 10b, it is shown that in the specimen formed by a laser power of 300 W and there were many hole defects in the fracture surface of the specimen formed by a laser power of 300 W. The cooling rate of the single melting point between different sedimentary hierarchies was different during the process of specimen forming, and the fluidity of liquid metal was affected. The surface quality of the specimen was poor, and the hole defects were formed. The hole defects caused stress concentration and the reduced tensile strength of the formed specimens. The specimen formed by a laser power of 250 W is shown in Figure 10c. Due to the lack of laser energy, the unmelted powder could be clearly observed, resulting in poor fusion between the deposited layers. When subjected to tensile force, the holes expanded into cracks, and tongue-shaped pits appeared on the fracture surface. Micro-holes appeared around the tongue-shaped pits, and the surrounding dimples were shallow, which greatly reduced the tensile strength of the formed specimen. Figure 10d shows the fracture surface of the 300 W–260 W uniform six-layer forming specimen. No obvious defects were found in the fracture surface, but there existed some micro-holes. The micro-holes had a lesser effect on the mechanical properties than the hole, and the fracture surface was relatively flat.

According to the analysis of mechanical properties and fracture morphology, there were many holes in the specimen formed by the QPFP. However, due to the decrease in volume energy density, the specimen formed by the VPFP makes up for the difference in the energy heat transmission rate between the energy deposition layer. The volume of the hole decreased and disappeared, and the stress concentration was greatly reduced. Thus, the mechanical properties of the specimen were improved. This indicates that the VPFP could improve the mechanical properties of the specimen.

### 3.3. Analysis of Microstructure

Figure 11 shows the optical microstructure of the tensile section of the specimen after polishing corrosion. Additionally, it presents that the microstructure of the tensile section of the TC4 titanium alloy specimen formed by SLM was mainly composed of the columnar crystal *β* phase and the acicular martensite *α’* phase. The acicular martensite *α’* phase in the columnar crystal *β* phase was interlaced and grew at the boundary of the columnar crystal *β* phase. The columnar crystal *β* phase of the specimen formed by laser power 300 W was coarse, and the width of the columnar crystal *β* phase of other parameters was not much different. With the increase in the number of uniform hierarchies shown in Figure 10b–d, the content of the acicular martensite *α’* phase decreased. In SLM, the rapid scanning of the laser beam and the rapid solidification of metal liquid led to a special Burgers relationship between the *α/α’* phase and *β* phase, the nucleation and preferential growth of the *α’* phase at the boundary of the columnar crystal *β* phase, and the growth along the direction of rapid cooling. There are obvious black domains in Figure 11. Lore Thijs et al. [25] pointed out that these black domains originate from the existence of some phases because the SLM process is a process of rapid cooling and solidification. At a certain temperature in this study, the solubility of Al in Ti was very low, and segregation and Ti3Al phase aggregation were prone to occur. These intermetallic compounds, such as Ti3Al, formed a black domain after corrosion. In addition, the spindles of these columnar crystals were perpendicular to the scanning direction and consistent with the superposition direction, resulting in different metallographic orientations within the organization and showing the phenomenon of light and dark alternation.

Figure 12 presents a six-layer (center and edge) SEM image of laser power 300 W and laser power 300 W–250 W. The SEM image of the specimen tensile fracture surface QPFP forming laser power 300 W and VPFP laser power 300 W–250 W divided into six hierarchies (the center and the edge of specimen). The size, morphology, and quantity of the acicular martensite *α’* phase in the QPFP was different from those in the VPFP. Additionally, the size of the acicular martensite *α’* phase was more uniform and the content of the secondary acicular martensite *α’* phase was more in the specimens formed by the QPFP with a laser power of 300 W. The laser power 300 W–250 W of the VPFP was divided into six hierarchies to form the specimen. The length of the acicular martensite *α’* phase along the forming direction was relatively reduced, the width was thicker, and the content of the secondary acicular martensite *α’* phase was reduced. Because in the process of VPFP forming, the energy density of the post-deposition layer was less than that of the pre-deposition layer and the cooling range was reduced, it was not enough to support the more formed secondary acicular martensite *α’* phase. This resulted in less of the secondary acicular martensite *α’* phase around the columnar crystal *β* phase. Comparing the SEM images of the center and the edge of the specimen, the *α’* phase grew along the direction of rapid cooling at the boundary of the columnar crystal *β* phase. When the specimen formed at the center, due to the high energy of the laser beam, the forming molten pool was higher than the powder laying height, resulting in the reverse growth of the martensite *α’* phase along the scanning direction of the forming layer. Additionally, the tensile section was perpendicular to the forming surface and the martensite *α’* phase presented a parallel state. According to the thermal conductivity of the powder, lower than that of the formed specimen during the forming process of the specimens, the heat dissipation grew along the direction of the formed specimen, and the martensite *α’* phase grew and was formed in this direction.

Through the analysis of the microstructure, the internal mechanism of the specimen formed by the VPFP was further revealed. The content of the secondary martensite *α’* phase of the specimen formed by the VPFP decreased. The length of the primary *α’* phase became shorter, the width increased with the superposition of hierarchies, and the width of the columnar crystal *β* phase became smaller than that of the columnar crystal *β* phase formed by the QPFP.

## 4. Conclusions

In this study, the tensile specimens with aerosolized TC4 titanium alloy powder were formed by SLM, using two different forming processes (the variable parameter forming process and the quantitative parameter forming process). Based on the test results, an investigation of the density, surface roughness, mechanical properties, fracture morphology, and metallographic structure of the specimens was carried out. The following major conclusions were obtained from this study:

1.The density of the specimen formed by the VPFP was higher than that of the specimen formed by the QPFP. The more hierarchies of the VPFP there were, the higher the density of the specimens was. Additionally, the highest density was 99.7%. The surface roughness could be improved by using a VPFP.2.The internal holes of the specimens formed by the VPFP were small in volume and occupied a relatively small proportion, which greatly reduced the phenomenon of stress concentration and improved the mechanical properties. The tensile strength of the specimen formed by a laser power of 300 W–260 W could reach 1185.214 MPa.3.The content of the secondary martensite *α’* phase decreased by the VPFP. With the superposition of hierarchy, the length of the primary *α’* phase became shorter, the width increased, and the width of the columnar crystal *β* phase was smaller than that of the QPFP.

## Figures and Tables

**Figure 1 materials-15-06408-f001:**
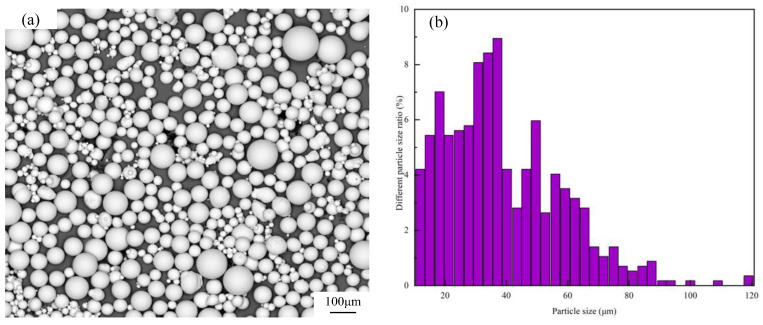
Microstructure and particle size distribution of TC4 powder. (**a**)Powder morphology; (**b**) Percentage of powder particle size.

**Figure 2 materials-15-06408-f002:**
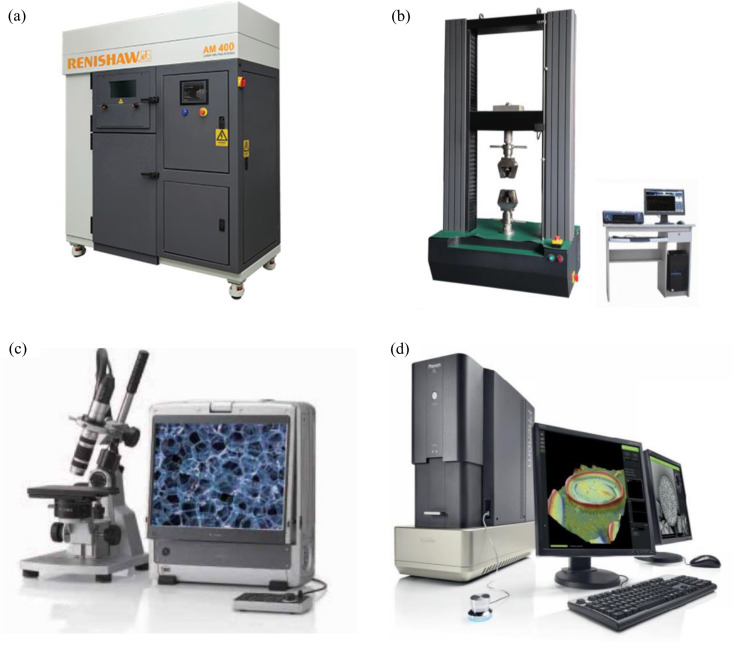
Experimental equipment. (**a**) Metal 3D printer; (**b**) Mechanical drawing machine; (**c**) Super deep scene 3D microscope; (**d**) Scanning electron microscope.

**Figure 3 materials-15-06408-f003:**
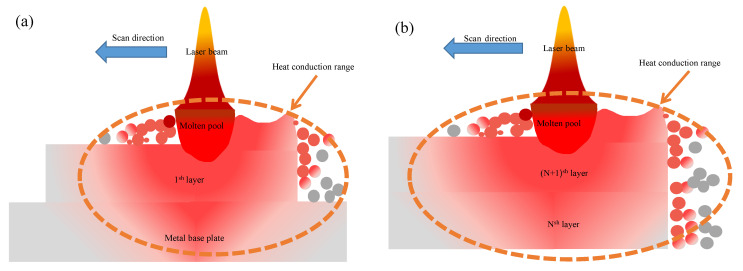
Temperature range of different molding heights. (**a**) Propagation range of metal base plate forming temperature; (**b**) The temperature propagation range of specimen-forming process.

**Figure 4 materials-15-06408-f004:**
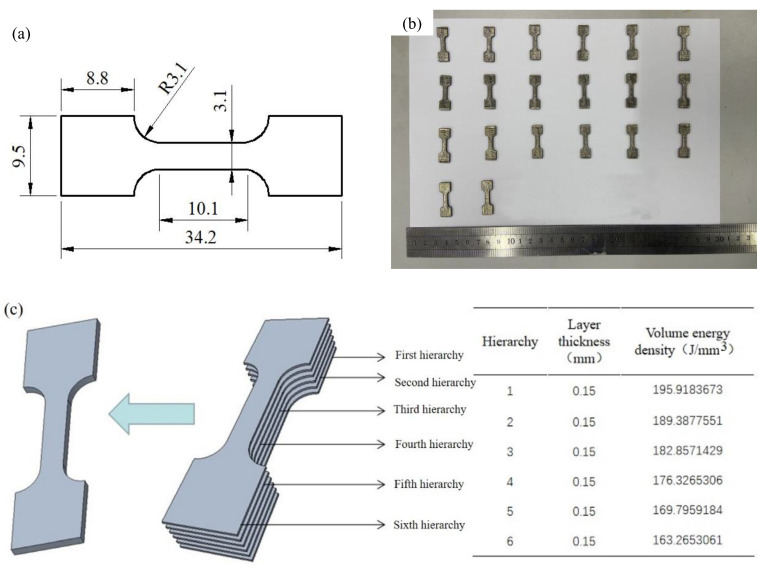
Size of tensile parts and distribution of physical density of six hierarchies of tensile parts. (**a**) Geometry of tensile parts; (**b**) Forming specimens; (**c**) Distribution of the physical density of six hierarchies of tensile parts.

**Figure 5 materials-15-06408-f005:**
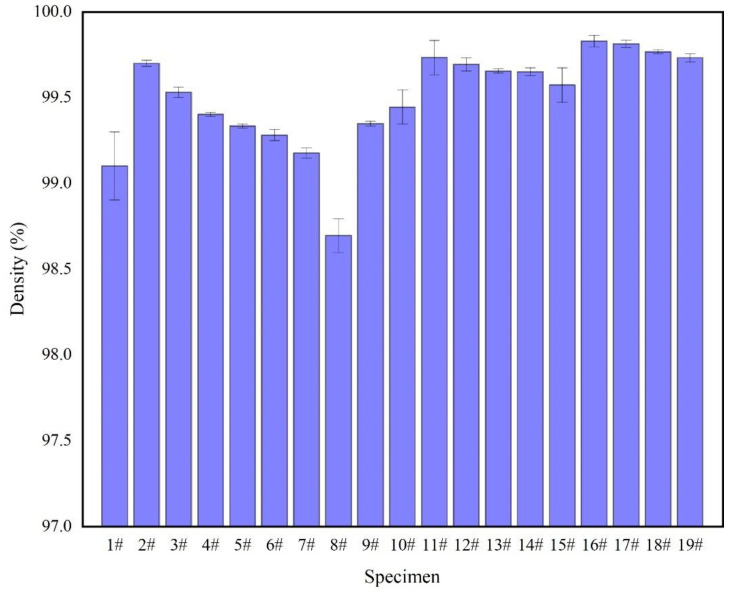
Density measurement results of specimen.

**Figure 6 materials-15-06408-f006:**
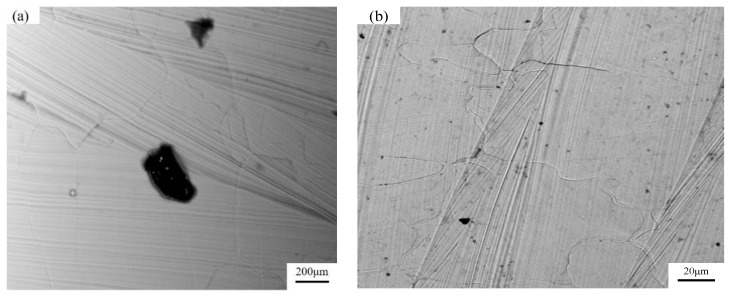
Auxiliary holes and cracks. (**a**) Auxiliary holes; (**b**) Cracks.

**Figure 7 materials-15-06408-f007:**
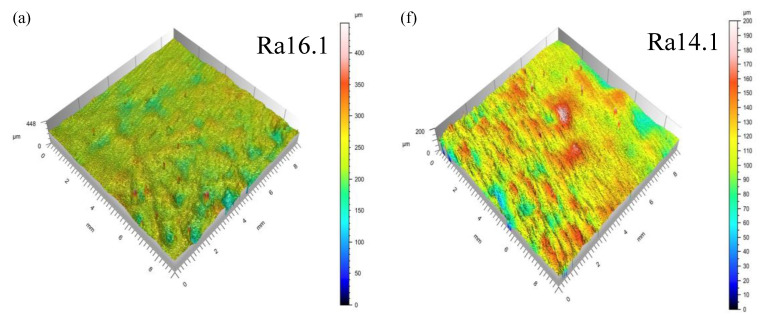
The surface roughness of the specimen. (**a**) 3#; (**b**) 4#; (**c**) 5#; (**d**) 6#; (**e**) 7#; (**f**) 12#; (**g**) 13#; (**h**) 14#; (**i**) 15#; (**j**) 9#.

**Figure 8 materials-15-06408-f008:**
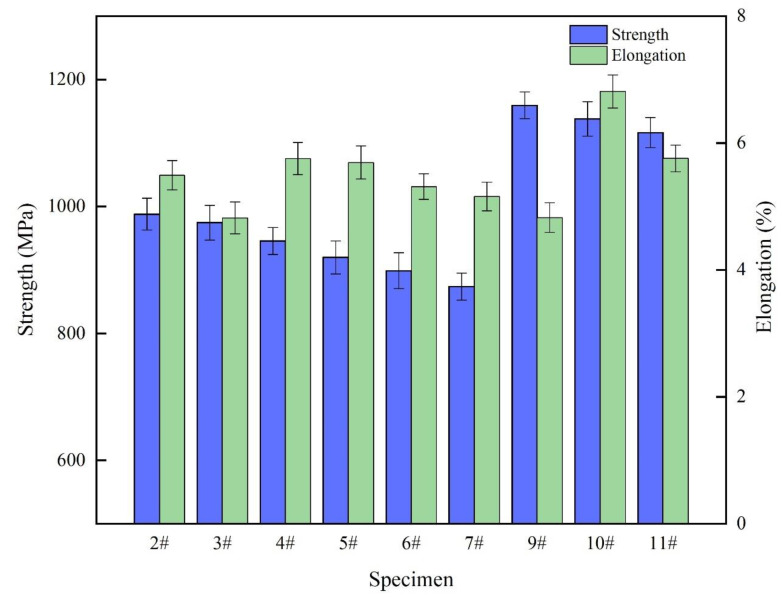
Tensile strength and elongation of the specimen formed by difference processes.

**Figure 9 materials-15-06408-f009:**
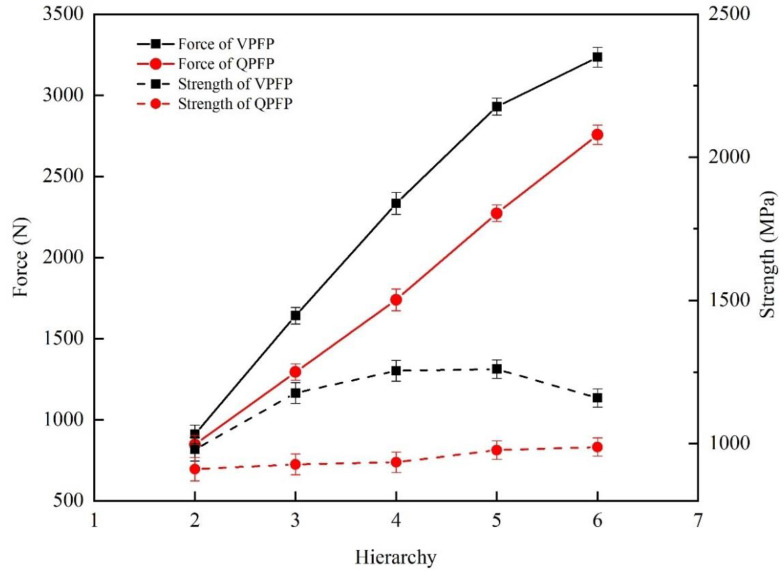
Trend diagram of tensile properties of specimens formed by laser power 300 W and laser power 300 W–250 W in six hierarchies.

**Figure 10 materials-15-06408-f010:**
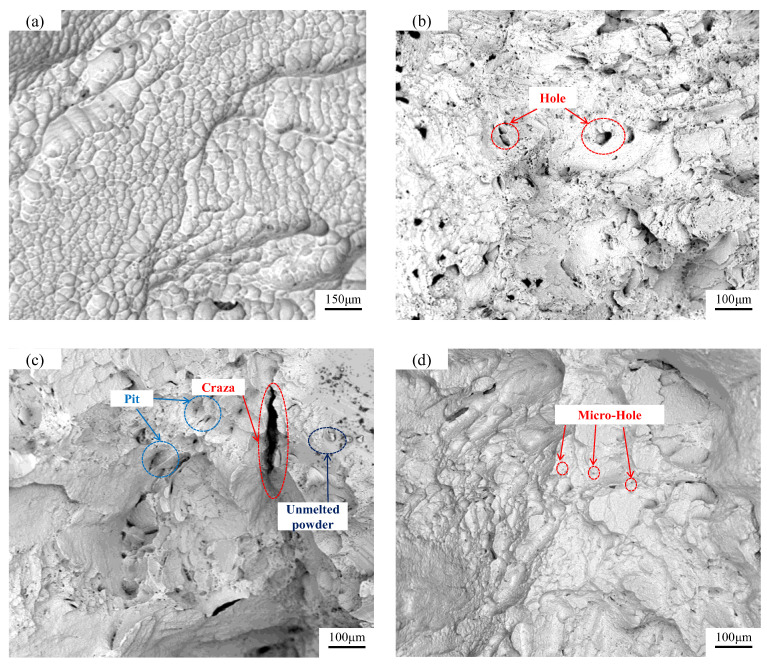
Tensile fracture morphology of specimens formed by different processes. (**a**) Morphology of dimple; (**b**) Laser power 300 W; (**c**) Laser power 250 W; (**d**) Laser power 300 W–260 W evenly divided into six hierarchies.

**Figure 11 materials-15-06408-f011:**
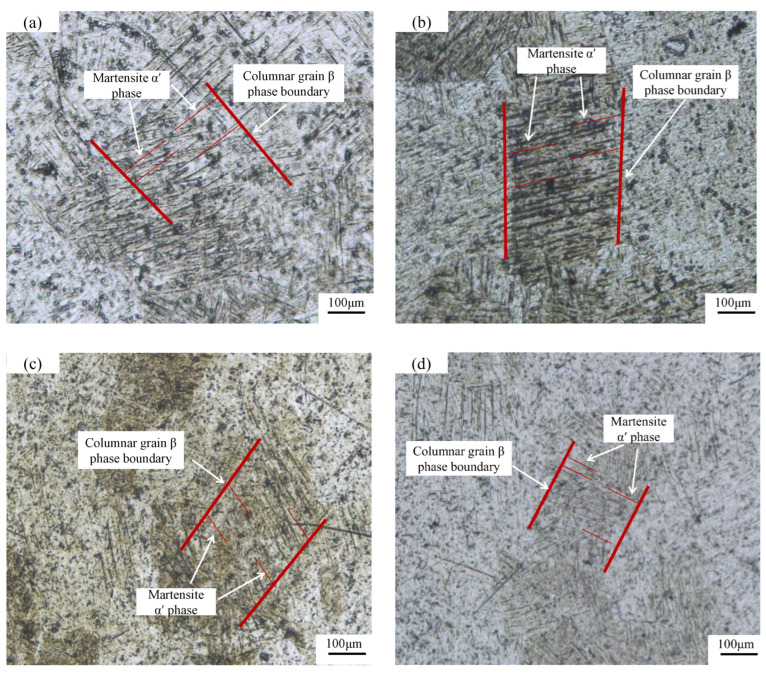
Metallographic structure. (**a**) 2#; (**b**) 11#; (**c**) 10#; (**d**) 9#.

**Figure 12 materials-15-06408-f012:**
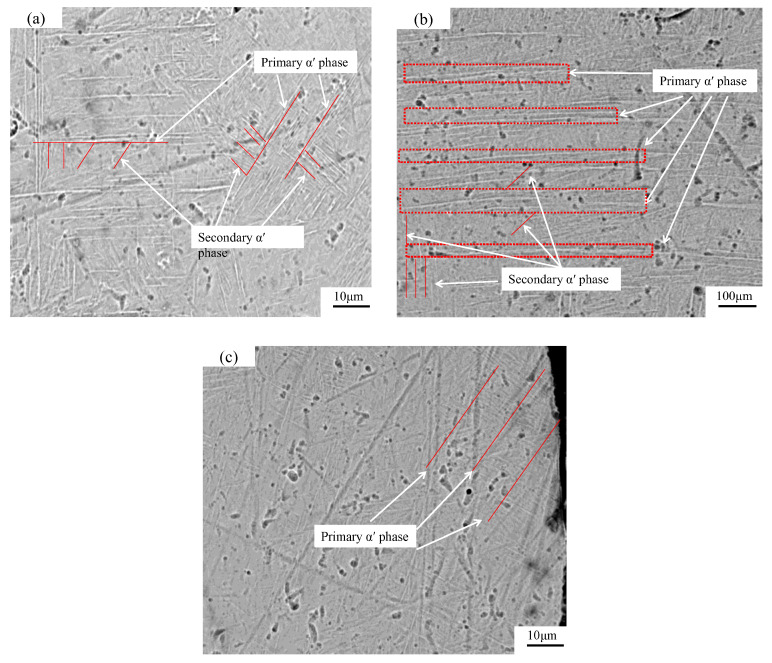
SEM image analysis. (**a**) 2#; (**b**) The center of specimen 9#; (**c**) The edge of specimen 9#.

**Table 1 materials-15-06408-t001:** Chemical compositions of the TC4 titanium alloy powder.

Element	Ti	Al	V	Fe	C	N	H	O
Wt.%	Balance	5.5~6.5	3.5~4.5	0.25	0.08	0.03	0.0125	0.13

**Table 2 materials-15-06408-t002:** Laser power and experimental parameters.

No.	Number of Hierarchy	Layer Thickness (mm)	Aggregate Thickness (mm)	Laser Power per Hierarchy (W)					
				1	2	3	4	5	6
1	1	0.9	0.9	350					
2	1	0.9	0.9	300					
3	1	0.9	0.9	290					
4	1	0.9	0.9	280					
5	1	0.9	0.9	270					
6	1	0.9	0.9	260					
7	1	0.9	0.9	250					
8	1	0.9	0.9	200					
9	6	0.15	0.9	300	290	280	270	260	250
10	3	0.3	0.9	300	275	250			
11	2	0.45	0.9	300	250				
12	2	0.15	0.3	300	290				
13	3	0.15	0.45	300	290	280			
14	4	0.15	0.6	300	290	280	270		
15	5	0.15	0.75	300	290	280	270	260	
16	2	0.15	0.3	300	300				
17	3	0.15	0.45	300	300	300			
18	4	0.15	0.6	300	300	300	300		
19	5	0.15	0.75	300	300	300	300	300	
20	6	0.15	0.9	300	392	284	276	268	260

## Data Availability

Data is contained within the article.

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
