# Peer review of "Experimental Research on Variable Parameter Forming Process for Forming Specimen of TC4 Titanium Alloy by Selective Laser Melting"

_materials, 2022, doi:10.3390/ma15186408_

Round 1

Reviewer 1 Report

The manuscript is aimed at TC4 alloy quality improvment which is not a new challenge but still important goal. The proposed approach allows to look at the problem from a different point of view and provided deep structural invstigations justify and support it. Thus, the manuscript should be interesting and helpful for readers. The advantage of this manuscript is that observed effects are discussed and explained from different points of view including density and defects, mechanical properties, fracture morphology, microstructure analysis.

General idea of the manuscript is that the influence of the heat transfer range of laser energy on the mechanical properties of the printed specimens is not usually considered, i.e. parts and samples are printed with fixed process parameters despite the different conditions during printing different layers and depending on distance to substrate. Thus, authors propose idea to vary process parameters with layers. In my opinion, this idea matters only in cases of thin samples oriented in XY plane printing. And the idea loses relevance in case of volumetric samples consist from many layers. It should be clear discussed in the manuscript in the end of Introduction section when the aim of the study is formulated (lines 67-81).

However, some explanations are strongly needed. Wang D et al [https://doi.org/10.3390/met8070471] in 2018 provided work also on Ti6Al4V and the experiment design is almost the same with presented in the manuscript. There are some differences, but overall that article and presented manuscript look very similar. Authors should refer to this article and explain novelty of proposed approaches and compare obtained results in details.

There are also some specific comments:

1)     Lines 88-91: The general information about TC4 alloy is unnecessary in Experimental procedure section.

2)     There is no information about material of the substrate used for SLM.

3)     As far as I understand all samples were printed in XY plane (direct information about it wasn’t found in the text), so what about support structures which affect on heat transfer significantly? In case of no supports were used and samples were melted directly on substrate information about cutting methods and substrate material is definitely needed.

4)     Lines 117-128: Discussion about temperature gradient, heat dissipation, etc. should be moved to Results and discussion section. And if authors discuss sample shrinkage (Line 121) corresponding measurements of the actual sample thickness would be interesting.

5)     Fig.2: In my opinion there is no need to show images of these experimental equipments, it is easy to find in the Internet by machine name given in text.

6)     Lines 142-147: More detailed explanation of the difference in process parameters for different hierarchies is needed. Based on which suggestions were laser power values given in Table 2 chosen?

7)     Fig. 7: There is no information how were surface profiles measured. It should be provided in Experimental section.

Author Response

Dear Editors and Reviewers:

Thank you for your letter and for the reviewers’ comments concerning our manuscript entitled “Experimental Research on Variable Parameter Forming Process for Forming Specimen of TC4 Titanium Alloy by Selective Laser Melting” (ID: materials-1778229). Those comments are all valuable and very helpful for revising and improving our paper, as well as the important guiding significance to our researches. We have studied comments carefully and have made correction which we hope meet with approval. The main corrections in the paper and the responds to the reviewer’s comments are as flowing:

Responds to the reviewer’s comments:

  1. Response to comment: (Lines 88-91: The general information about TC4 alloy is unnecessary in Experimental procedure section.)

Response: Thank you for your guidance. TC4 information has been deleted

  1. Response to comment: (There is no information about material of the substrate used for SLM.)

Response: We are very sorry for our negligence of the information about material of the substrate used for SLM. We have added relevant information

  1. Response to comment: (As far as I understand all samples were printed in XY plane (direct information about it wasn’t found in the text), so what about support structures which affect on heat transfer significantly? In case of no supports were used and samples were melted directly on substrate information about cutting methods and substrate material is definitely needed.)

Response: Thank you for your guidance. This experiment has a support structure, and relevant information has been added to the article.

  1. Response to comment: (Lines 117-128: Discussion about temperature gradient, heat dissipation, etc. should be moved to Results and discussion section. And if authors discuss sample shrinkage (Line 121) corresponding measurements of the actual sample thickness would be interesting.)

Response: Thank you for your valuable suggestions. This part is mainly based on the rules found by summarizing other papers. This experiment is designed based on these rules, and compared with common SLM process forming by some characterization methods, so as to draw relevant conclusions.

  1. Response to comment: (Fig.2: In my opinion there is no need to show images of these experimental equipments, it is easy to find in the Internet by machine name given in text.)

Response: Thank you for your valuable advice. Fig.2 can provide readers with more intuitive reading experience and facilitate them to learn more information about the machine when reading. 

  1. Response to comment: (Lines 142-147: More detailed explanation of the difference in process parameters for different hierarchies is needed. Based on which suggestions were laser power values given in Table 2 chosen?)

Response: We gratefully appreciate for your valuable suggestion, related content has been added.

  1. Response to comment: (Fig. 7: There is no information how were surface profiles measured. It should be provided in Experimental section.)

Response: Thank you so much for your careful check, related content has been added.

Reviewer 2 Report

A few points are given below

1. What is the use of Fig3? Its not clear 

2. What is hierarchy ? Its use is not clear

3. Table 2 does not look like Number of specimens. It looks like specimen no.

4.What is aggregate thickness ?

5. What is floor ? This is not common AM language

6.Quality of Fig 6 is extremely low. The polishing is not done properly

7.Use of fig 9 is unclear

8.Fig10 doe snot show proper images. These type of images have never been reported. The authors are requested to re-check their work

Author Response

Dear Editors and Reviewers:

    Thank you for your letter and for the reviewers’ comments concerning our manuscript entitled “Experimental Research on Variable Parameter Forming Process for Forming Specimen of TC4 Titanium Alloy by Selective Laser Melting” (ID: materials-1778229). Those comments are all valuable and very helpful for revising and improving our paper, as well as the important guiding significance to our researches. We have studied comments carefully and have made correction which we hope meet with approval. The main corrections in the paper and the responds to the reviewer’s comments are as flowing:

Responds to the reviewer’s comments:

  1. Response to comment: (What is the use of Fig3? Its not clear)

Response: Thank you for your guidance. The main function of Figure 3 is summarized from relevant papers, which is the original intention of this experiment.

  1. Response to comment: (What is hierarchy? Its use is not clear)

Response: Thank you for your guidance, in this paper, the hierarchy refers to the number of layers divided, this design will be formed specimen equal division.

  1. Response to comment: (Table 2 does not look like Number of specimens. It looks like specimen no.)

Response: Thank you for your guidance, We have made improvements.

  1. Response to comment: (What is aggregate thickness.)

Response: Thank you for your guidance, Aggregate thickness is the thickness of hierarchy.

  1. Response to comment: (What is floor? This is not common AM language)

Response: Thank you for your careful inspection. Floor has been changed to hierarchy

  1. Response to comment: (Quality of Fig 6 is extremely low. The polishing is not done properly)

Response: Thank you for your guidance, the figure has been optimized

  1. Response to comment: (Use of fig 9 is unclear)

Response: The figure has been optimized and further illustrated

  1. Response to comment: (Fig10 doe snot show proper images. These type of images have never been reported. The authors are requested to re-check their work)

Response: Thank you for your guidance, Fig10 showed that the defact of the tensile fracture morphology of specimens.

Round 2

Reviewer 1 Report

Authors response and corrections in the manuscript are connected only with list of specific comments and there no response to general comments about relevance of the obtained results for volumetric samples and comparison with previous similar studies (for exumple, Wang D et al [https://doi.org/10.3390/met8070471]).

Thus, in my opinion, Introduction section needs significant revision.

Moreover, I'm not satisfied with some given answers. Namely, line 142: "The data of this experiment is the selection of QPFP, and the VPFP is carried out on 142 the basis of the existing experiment".

The phrase "based on existing experiment" makes no sence. It must either be described in details, or a relevant reference on previous authors' study is needed.

Thus, still major revision is needed to make the manuscript appropriate for Materials jounal scientific standards. 

Author Response

Dear Editors and Reviewers:

Thank you for your letter and for the reviewers’ comments concerning our manuscript entitled “Experimental Research on Variable Parameter Forming Process for Forming Specimen of TC4 Titanium Alloy by Selective Laser Melting” (ID: materials-1778229).

First of all, I'm really sorry that I didn't reply to some of your instructions in the last reply process. Thank you for your guidance on this paper again, so that this paper can be improved and more relevant knowledge can be learned.

Responds to the reviewer’s comments:

1.Response to comment: (The idea loses relevance in case of volumetric samples consist from many layers. It should be clear discussed in the manuscript in the end of Introduction section when the aim of the study is formulated (lines 67-81).)

Response: Thank you for your guidance. Related content has been added in the introduction section (in the red section of the article).

2.Response to comment: (Authors response and corrections in the manuscript are connected only with list of specific comments and there no response to general comments about relevance of the obtained results for volumetric samples and comparison with previous similar studies (for exumple, Wang D et al [https://doi.org/10.3390/met8070471])

Response: Thank you for your guidance. Wang D et al. studied the full-factor experiment was performed to obtain the optimal combination of parameters including laser power, scanning speed, layer thickness, scanning spacing, and laser spot size. Modifiability of these parameters was considered jointly, and the influence of each factor on SLM fabrication quality analyzed. And the result that the SLM-fabricated titanium alloy into five zones: severe over-melting zone, high-energy density nodulizing zone, smooth forming zone, low-energy density nodulizing zone, and sintering zone. In this experiment, the formed specimen was stratified and the mechanical properties of the specimen were improved through a new process. Although the characterization means were similar in the experimental process, the design process for the test was completely different.  As for the results, in this experiment, compared with the quantitative parameter forming process (QPFP), the surface morphology, tensile properties, and microstructure of the specimens was improved by the variable parameter forming process (VPFP). Wang D et al mainly discussed the different types of parts manufactured by SLM, the parameter window of smooth forming zone was explored, and the mechanism of improving ductility of titanium alloy prepared by SLM.

3.Response to comment: (The phrase "based on existing experiment" makes no sence. It must either be described in details, or a relevant reference on previous authors' study is needed.)

Response: Thank you for your guidance. Related references and detailed instructions have been added in the red section of the article.

Finally, thank you again for your careful guidance, let us benefit a lot.

Round 3

Reviewer 1 Report

I am satisfied with given answers, corrections and explanations. There is some uncertainty concerning scientific significance and novelty, but this could be left to the reader's judgment.

I would recommend the manuscript to being accepted in the present form.

Author Response

Thank you for your approval of this article.